# Hepatitis E Virus Detection in Hunted Wild Boar Liver and Muscle Tissues in Central Italy

**DOI:** 10.3390/microorganisms10081628

**Published:** 2022-08-11

**Authors:** Gianluigi Ferri, Carlotta Lauteri, Anna Rita Festino, Andrea Piccinini, Alberto Olivastri, Alberto Vergara

**Affiliations:** 1Faculty of Veterinary Medicine, Post-Graduate Specialization School in Food Inspection “G. Tiecco”, University of Teramo, Strada Provinciale 18, Piano d’Accio, 64100 Teramo, Italy; 2Veterinary Service I.A.O.A., ASUR Marche, Area Vasta 5 Ascoli Piceno/San Benedetto del Tronto, 63900 Fermo, Italy

**Keywords:** hepatitis E virus, RNA, wild boar, liver, muscle, food-chain, one-health

## Abstract

In different European countries, including Italy, hepatitis E virus (HEV) has been recognized as an emerging public health concern. Humans are infected through the orofecal route by the ingestion of contaminated uncooked or undercooked animal-origin foodstuffs. Wild boars (*Sus scrofa*) have gained a crucial role as viral reservoirs. HEV-3 is the most frequently identified genotype from hunted wild boar liver and muscle tissues. The Marche region, more specifically Ascoli Piceno province, is characterized by a rooted hunting tradition and related product consumption. In this research study, 312 liver and 296 muscle specimens were screened using biomolecular assays, and HEV RNA was detected from 5.45% and 1.35% of liver and muscle samples, respectively. Phylogenetic analysis revealed that positive animals were infected by genotype 3 subtype c. Based on the environmental pathogen characteristics, HEV has also evolved to guarantee its survival in a wild environment. Therefore, wild boars and ruminants have a key role in its persistence. Epidemiological data regarding HEV circulation have resulted as necessary, and biomolecular analysis represents an important means of monitoring and establishing preventive measures. A multidisciplinary approach could provide a wide perspective regarding HEV and infectious implications on human, animal, and environmental health.

## 1. Introduction

Viral pathogens have acquired relevant scientific recognition as crucial issues in certain food production chains, which have included industrial and artisanal production systems [1]. This consideration has highlighted the importance to consider further microbiological monitoring activities to guarantee safe alimentary matrices to the final consumer. In particular, the so-called “Good Hygiene Practices” (reported in the EU Reg. No. 852/2004) had already anticipated the importance for food operators and competent authorities to apply and control, respectively, sanitary measure improvements at the production level. This European Regulation has highlighted their critical role in order to prevent high biological risks, i.e., infection caused by zoonotic viral microorganisms [2].

Slaughterhouses represent a particular environment in which viruses and bacteria can easily come in contact with humans, providing favorable conditions to enforce their persistence [3].

Every year, hepatitis E virus (HEV), due to its zoonotic transmission, causes numerous infections and has also been classified as a professional illness due to the high levels of seroprevalence data discovered in workers’ sera, i.e., veterinary officials, slaughterhouse personnel, hunters, etc. [4].

Generally, HEV infects the human host through the ingestion of contaminated uncooked or undercooked animal-origin foodstuffs and waters [5], but it has also been confirmed that direct contact with infected subjects is responsible for seroconversion, too [3]. HEV is classified as a non-enveloped single-strand RNA virus that belongs to the *Hepeviridae* family [6]. It presents three overlapping open reading frames (ORF): ORF1, 2, and 3. From a physio-pathological point of view, ORF2 encodes for a strategical capsid protein, which permits the cyto-receptor interaction with specific host cytotypes (more expressed by enterocytes, hepatocytes, and myocytes). The involved structure is represented by inter-membranal complexes named heparan sulfate receptors that have a key role for infection [7]. Due to its structural characteristics, HEV can persist in the environment for several days [8]. In this way, in cases of environmental sharing between wild and domestic animal hosts, HEV can persist and consequently diffuse in two viral life cycles: “urban” and “wild” ones. It determines repercussions on consumers’ health [9,10].

Nowadays, researchers have identified eight different genotypes (from HEV-1 to HEV-8) from different mammalian hosts. HEV-3 and 4 have been mostly reported (identified from humans, domestic pigs, and wild boars) in industrialized countries, while the other ones have been mainly reported from developing countries [1,2,3,11].

Depending on HEV genotype, generally human patients have resulted as asymptomatic or pauci-symptomatic (HEV-3 4); on the contrary, HEV1-2, which has been more frequently identified in developing countries, induces severe symptoms such as diarrhea, vomiting, etc. [3].

HEV-3 has been largely identified in many European countries, including Italian regions [12]. Domestic swine has represented the main HEV-3 reservoir; wild boar gains an important role as a viral wild reservoir (and also wild ruminants) [13]. Indeed, many authors justified wild boar positivity to several factors: habitat sharing between domestic and wild animal species, swine farming techniques (extensive or intensive husbandries), domestic and wild animal densities and geographical characteristics regarding the screened area [13].

The aim of this study was to provide data regarding HEV genotype and subtype circulation in wild boars from low-screened geographic regions: the Ascoli Piceno province and the Southern Marche region (characterized by a rooted wild boar hunting and consumption tradition). The lack of information about HEV RNA from muscle samples, which can impact the emerging food production chains, has induced the necessity to conduct a further investigation.

Therefore, two parallel biomolecular screenings involving 312 liver and 296 muscle tissue samples collected from hunted wild boars (*Sus scrofa*) were combined in the present research article. This research article has been based on a comparative and multidisciplinary approach and evaluated possible sanitary repercussions on the final consumer’s health.

## 2. Materials and Methods

### 2.1. Samples Collection

During the hunting season 2019/2020, an amount of 312 wild boars (*Sus scrofa*) were molecularly screened for HEV RNA detection. Three-hundred and twelve liver and two-hundred and ninety-six muscle tissue (diaphragm) samples were collected from provided plucks and muscle samples at a slaughterhouse in central Italy (Ascoli Piceno province), in accordance with the Regional Marche Law No. 3/2012, which requires plucks’ *postmortem* evaluation of hunted wild boars. The sanitary control activities that are performed by the Veterinary Public Services (EU Reg. No. 625/2017 and EU Reg. No. 624/2019) also included *Trichinella* spp. surveillance (in accordance with EU Reg. No. 1375/2015).

More detailed characteristics regarding the screened population are illustrated in Table 1.

The screened area included 10,251 ha characterized by the following wild fauna density data: wild boars 2.5/100 ha and wild ruminants < 1/100 ha. This province was characterized by a low pig farm density (as intensive farming systems), small rural communities with one or two pigs/family for domestic consumption (as extensive techniques), low anthropization levels, and numerous mountain areas with reduced logistic access to humans.

For each sample type, 25 g aliquots were collected and sterilely sampled, avoiding cross-contaminations from serosa to the parenchyma, as reported by Dzierzon et al. [14]. Successively, all samples were transported under refrigerated conditions and stored at −80 °C until their biomolecular screenings.

### 2.2. RNA Extraction and Nested RT-PCR Assays

The first sample processing step included liver and muscle tissue homogenization performed using T18 digital Ultra-Turrax^®^. Successively, HEV RNA was extracted through the TRIzol LS method (Invitrogen, Ltd., Paisley, UK) and molecularly screened with nested RT-PCR assay.

The first molecular reaction (RT-PCR) was performed using Qiagen^®^ (Hilden, Germany) OneStep RT-PCR Kit and Green Master Mix Promega^®^ (Madison, WI, USA) for the second one (nested PCR). All reactions were performed in a total volume of 25 μL by using specific primers for HEV RNA detection targeting regions belonging to the ORF1 and ORF2 genes [15], as illustrated in Table 2. Amplicons were successively loaded on the agarose gel and nitid bands were compared to specific DNA ladders (Genetics, FastGene^®^ 50 bp or 100 bp DNA Marker) [16].

### 2.3. Sequencing and Phylogenetic Analysis

The obtained amplicons, which resulted as positive to the molecular screening, were purified through the usage of Qiagen QIAquick^®^ PCR Purification Kit. These purified samples were successively sequenced by BioFab Research (Rome, Italy).

The nucleotide similarity was performed by using the BLAST system (http://www.ncbi.nlm.nih.gov/genbank/index.html, accessed on 30 March 2020).

Sequence alignments and evolutionary analysis were conducted in the MEGA X software [17] and the phylogenetic tree was constructed using the Neighbor-Joining method [18] (See Figure 1). Positive sequences were deposited in the GenBank database and published.

### 2.4. Statistical Analysis

Regarding HEV RNA prevalence and relative percentage values, confidential intervals (CI) were calculated. The XLSTAT 2014 software^®^ (Renmond, Washington, DC, USA) was used to provide correlation by calculating the chi-square statistic value (with Yates’s correction) where it is applicable. The Pearson correlation (r) coefficient analysis was calculated between two parameters, and findings of *p* < 0.05 were considered significant.

## 3. Results

The biomolecular screening, performed in order to detect HEV RNA from hunted wild boar tissues during the season 2019/2020, showed the following positivity rates: 17/312 (5.45%, 95% CI: 1.7–8.9%) liver and 4/296 (1.35%, 95% CI: 1.1–2.5%) muscle samples presented ORF2 amplicons (145 bp).

All positive muscle samples also presented HEV RNA in their respective liver specimens, and none of the screened animals were positive from the muscle only. Focusing on muscular tissue positivity in more detail, the ORF2 amplicons were observed in diaphragm specimens both in female and male infected subjects (See Table 3).

As reported in Table 3 and Table 4, adult male muscle tissues resulted as significantly more positive than female ones. Sex- and epidemiological-based classification are explained in the following Table 4. These last two factors showed a significant relationship by using the chi-square statistic value (χ^2^: 22.3284). The related *p* value was <0.00001.

Among liver samples, female specimens presented a significantly higher positivity to the ORF2 amplicons than male ones. More specifically, in both screened genders, adults were significantly more represented than puberal animals. None of the screened subjects belonging to the juvenile age category resulted positive for HEV RNA detection.

All positive animals were hunted from different parts belonging to the Ascoli Piceno province, but included in an area range between 5 and 20 km. More detailed information is illustrated in Table 3 and Figure 1.

Nucleotide sequence evaluation of positive samples was performed by using the BLAST system and showed high similarities (98.0% nt identity) to the HEV genotype 3. The phylogenetic analysis demonstrated that all sequenced specimens belonged to the subtype c. They were registered and published on GenBank (https://www.ncbi.nlm.nih.gov/genbank/, accessed on 15 May 2022) with the following accession numbers: ON364349, ON364350, ON364351 and ON364352 (see Figure 2).

## 4. Discussion

In different Italian regions (including Marche), there is a rooted wild boar hunting tradition, and related foodstuff consumption determines an improvement in the hygienic game food production chain [9]. This trend has also been supported by a significant wild boar demographic increase, as reported by the Public Veterinary Services [19]. The consequential primary- and processed-food production chains have conduced researchers to pose scientific questions about sanitary and hygienic implications for human, animal, and environmental health [2].

HEV RNA detection, identified from wild boar liver samples, has been considered a previous marker for potential viral carcass contamination [5]. However, it is also mandatory to consider that the HEV RNA detection, exclusively located in the liver tissue, cannot be representative of a viremic status, as demonstrated by Risalde et al. [20]. They observed, through biomolecular and immunohistochemical screening, HEV persistence in liver samples collected from non-viremic, naturally infected wild boars. It confirmed a fundamental aspect that hepatocytes remain the target cells for massive viral multiplication in accordance with pathogen physiopathology [3,7]. Indeed, the hepatic biomolecular screenings can give precious epidemiological information about HEV genotypes’ and subtypes’ circulation in wild boar populations (including at the slaughterhouse level) [11,21]. For these reasons, HEV RNA detection only from liver samples cannot be synonymous for viremic animals and cannot permit us to exclude any sanitary repercussions; therefore, a parallel muscular screening should be necessary. Indeed, many countries have included HEV in the screened pathogens in order to guarantee safe products for the final consumer [3].

In this research study, in all positive animals, HEV-3 subtype c was detected as one of the most frequently identified in central Italy, as previously observed [11,12,13,22,23,24,25].

The obtained HEV RNA liver prevalence of 5.45% (95% CI: 1.7–8.9%) was in line with values reported by a preliminary study conducted in the same province of 5.12%, as reported by Ferri et al. [11], in which HEV3 subtype c was discovered. The trends observed in other Italian regions were: 3.7% in Toscana [26], 1.9% in Liguria [27], and 1.2% in Lombardy [13]. Conversely, these data were in contrast with an HEV RNA prevalence value of 9.0% reported by Aprea et al. [23] in the Abruzzo region. Furthermore, this last percentage was lower than the previous prevalence of 13.7% evidenced by the same research group [22] and reported in the same geographical area, but in the hunting season of 2018. These fascinating trends highlighted the pathogen’s pleomorphic distributions and its epidemiology across years. Indeed, in 2018, scientists primarily identified HEV3 subtype c, in contrast with 2020 data, when the same authors reported three different circulating subtypes, c, e, and f. These differences are directly linked to the wild boar demographic increase and underline the importance of monitoring activities regarding HEV3 subtypes’ circulation over the years [9].

In this study, positive muscle tissues were lower than hepatic ones. This trend has also been reported by other studies [12,21,23]. In this screening, 1.35% (95% CI: 1.1–2.5%) of diaphragm samples presented HEV RNA. In none of the cases were targeting regions (both ORF1 and ORF2) from muscle only observed, as described by De Sabato et al. [12]. In their investigation, HEV RNA was detected from 4.6% of screened aliquots. Generally, muscular viral detection could be justified by two potential reasons: cross-contaminations due to the anatomical contiguity between liver and diaphragm (including organ manipulations) or to the animal viremic status [9]. The real risk of cross-contamination has been recently and practically demonstrated by Dzierzon et al. [14] at the slaughterhouse level in the domestic swine model. It permitted us to support the consideration that anatomic contiguity and same containers with different livers (coming from different animals) resulted in the ability to provide perfect conditions for HEV cross-diffusion. Consequently, to these considerations, muscle tissue biomolecular screenings could reveal the viremia phase and potential risks for the final consumer’s health through avoiding the carcass’s introduction to the food chain (including familial consumption) [7].

In our study, to avoid the above-mentioned conditions, liver and muscular aliquots were sterilely sampled avoiding cross-contamination vectored by operators during cutting from serosa to the parenchyma, as observed by Dzierzon et al. [14]. Excluding this last factor, as a possible viral driver, our scientific hypothesis permitted us to suggest that positivity to the HEV RNA was linked to a possible viremic status [7,12,14], although this could not be confirmed due to the fact that blood was not available for HEV RNA detection.

In general, the prevalence values, obtained from liver and muscle samples, were justified by viral environmental persistence and circulation among the screened geographical areas. This meant that animal infections occurred from wild infected feces that diffused HEV [23]. In more detail, Ascoli Piceno province was characterized by low wild fauna densities (less than 2.5 wild boars/100 ha and <1 wild ruminants/100 ha). Associating this last observation to the ethological wild boar peculiarities and to the viral environmental behavior (high resistance of viral particles) justified the registered positivity in small animal numbers living in a range of 20–50 km, as observed by Arnaboldi et al. [13] in Sondrio province. In both cases, the screened areas presented different common characteristics: low anthropization levels, low domestic swine conventionally and extensively farmed (avoiding habitat sharing with wild animals and consequently cross-species transmission), and no fields resulted to be fertilized with pig manures as potential source for viral diffusion (as potential HEV source).

Therefore, positive cases can be justified by the HEV environmental persistence [10] in these specific areas; indeed, as previously reported by Aprea et al. [23], positive animals obtained from this study came from the same kilometrical range observed in our previous research [11]. This enforces the possibility of environmental infection enforcement and underlines the evolutionary strategy that HEV had developed in order to find favorable hosts for its survival and diffusion [3].

Finally, basing the dissertation on animal gender, in the present investigation adult females were significantly more positive than male ones (as reported in Table 3: Section 3). The same pattern was also observed by other research studies [13,21,23]. This difference can be explained by the hormone actions on heparan sulfate expression. Estrogens induce a positive signal, which improves heparan sulfate gene encoding. This condition confers to the enterocytes, hepatocytes, and myocytes a major receptiveness to ORF2 capsid protein expressed by HEV. In this way, mammalian hosts are more likely to be infected, simplifying viral multiplication [28]. From a comparative point of view, this aspect was also observed in a human host during acute infection. In this case report, in a Spanish hospital, the Infectious Disease Unit identified HEV RNA and high immunoglobulin titers from human breast milk, suggesting estrogens could significantly increase viral multiplication [29]. These considerations need to be furtherly supported by other research investigations, but it is clear that female hormones have interesting roles as inductive signals increasing host receptiveness.

Finally, the present study was constructed as a qualitative investigation in order to provide further epidemiological information regarding HEV genotype circulation in a particular Italian region characterized by a rooted hunting tradition and game-food-derived product consumption. The inclusion of muscle samples in this screening wanted to highlight a reasonable risk for consumer health, but this risk will be furtherly confirmed by the introduction of future quantitative biomolecular analysis (real-time Rt-PCR). This scientific necessity is based on providing information about viral copies per gram of tissue. It permits us to evaluate data and to calculate risks for the final consumers and guarantee safe products for immunocompetent and immunodeficient human subjects, as reported by Rivero-Juarez et al. [30]. In this way, public health institutions have a fundamental role in the improvement of food producers’ education about biological risks, especially focusing on emerging pathogens, i.e., HEV. Furthermore, the American and Canadian Sanitary Authorities indicated implementation of food labels (*Suidae* origin products) with information regarding HEV infection risks in order to prevent potential infections [31]. This study and similar ones pose further attention to this emerging pathogen, highlighting that the improvement of surveillance programs will be strictly necessary.

A One-Health approach is mandatory to better understand the “dark matter” of this pathogen, especially the sanitary impact on different food processing technologies. Biomolecular screenings have a crucial role in monitoring activities and possibly to prevent the entrance of carcasses and liver products into the game food chain. It also must be supported by more attention by food operators during evisceration processes and organ manipulations, as cross-species infection through direct contact (from animals to humans and vice versa) is possible. These measures will be strictly necessary to guarantee all consumers’ health. Therefore, starting from these observations, the multidisciplinary approach based on a comparative point of view can provide solid fundamentals for further scientific investigations.

## 5. Conclusions

In conclusion, due to the hunting and food-producing traditions and ecological characteristics of the screened area, HEV has survived across the years.

This was possible because in this geographical area there is a large consumption of primary and processed wild boar meat and liver products. Similar peculiarities have been observed in other Mediterranean countries, i.e., Spain and France [9].

Therefore, molecular biology has a strategic role in the epidemiological aspects and liver and muscle tissue analysis that could represent significant preventive medicine measures to avoid their entrance in food preparation processes. To prevent infections, important health education for worker categories at risk (i.e., veterinarians, hunters, food operators, slaughterhouse workers, etc.) and the final consumer is also required.

## Figures and Tables

**Figure 1 microorganisms-10-01628-f001:**
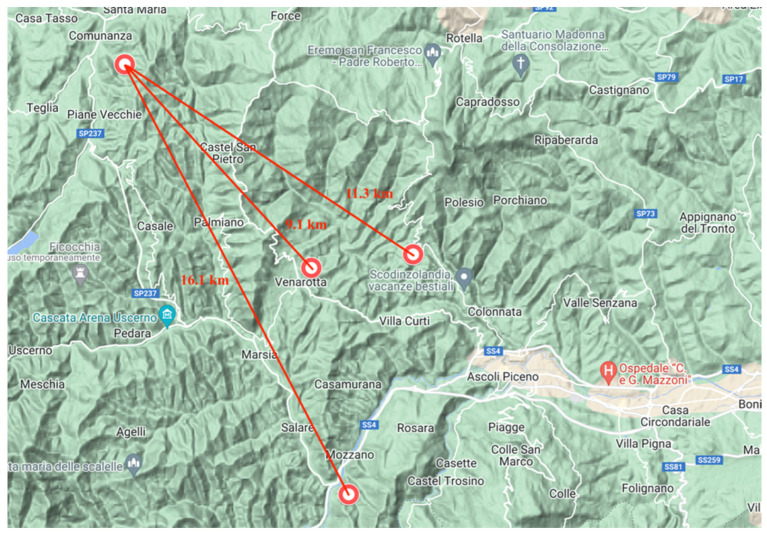
Positive animal samples: hunting localization. Red circles identify where animals were hunted.

**Figure 2 microorganisms-10-01628-f002:**
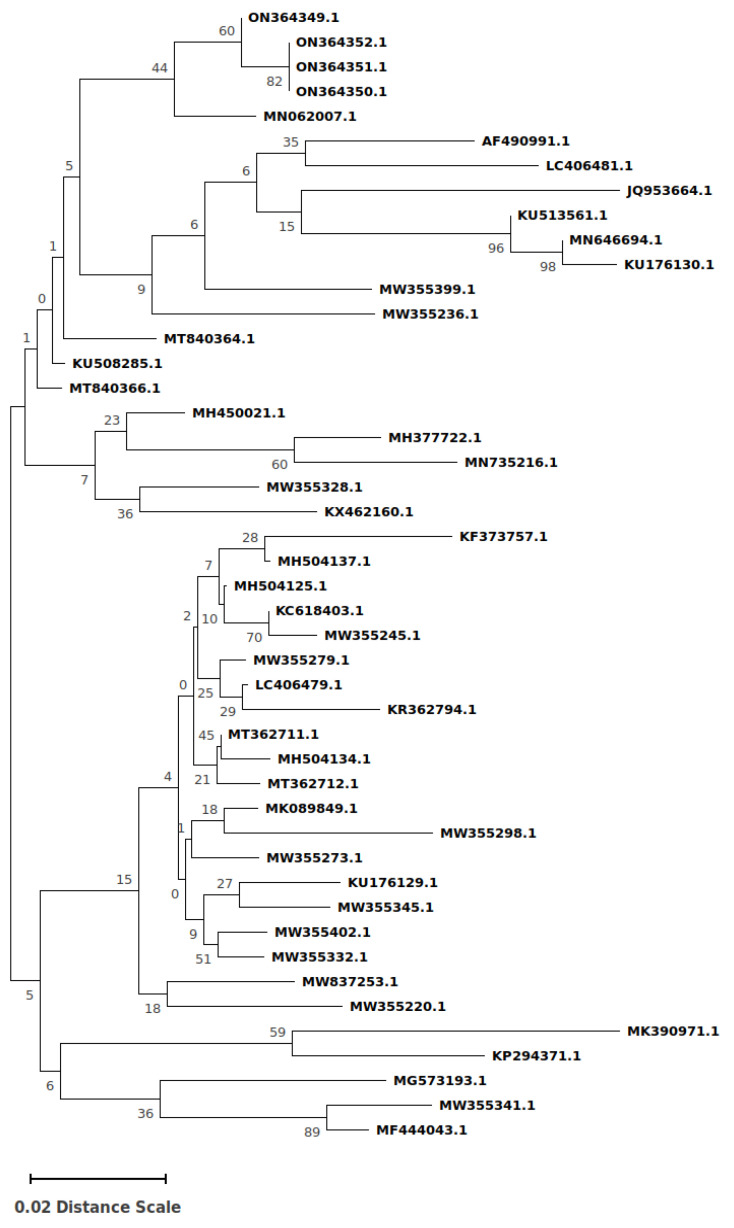
Neighbor-joining phylogenetic tree was constructed on the fragment of 145 bp, referring to the ORF2 from liver and muscle tissue samples and realized through the usage of p-distance model with bootstrapping of 1000 replicates. Sequence ON364349 was identified from muscle tissue (diaphragm) and the other ones ON364350, ON364351 and ON364352 from liver aliquots.

**Table 1 microorganisms-10-01628-t001:** Screened population for HEV RNA detection.

Sex	Age-Based Classification
148 M	3 J 41 P 104 A
164 F	2 J 57 P 105 A

M: Male; F: Female. J: Juvenile (weight < 15 kg and estimated age between 0–12 months). P: Puberal (15 kg < weight < 40 kg, and estimated age between 13–24 months). A: Adult (weight > 40 kg and estimated age between 24–48 months).

**Table 2 microorganisms-10-01628-t002:** Target genes used for HEV RNA detection, as reported by Wang et al. [15].

Genes	PCR Reactions	Primers	Oligonucleotide Sequences	Amplicon Size (bp)
**ORF1**	RT-PCR	ConsORF1-s1	F: CTGGCATYACTACTGCYATTGAGC	418 bp
		ConsORF1-a1	R: CCATCRARRCAGTAAGTGCGGTC	
	Nested PCR	ConsORF1-s2	F: CTGCCYTKGCGAATGCTGTGG	287 bp
		ConsORF1-a2	R: GGCAGWRTACCARCGCTGAACATC	
**ORF2**	RT-PCR	ConsORF2-s1	F: GACAGAATTRATTTCGTCGGCTGG	197 bp
		ConsORF2-a1	R: CTTGTTCRTGYTGGTTRTCATAATC	
	Nested PCR	ConsORF2-s2	F: GTYGTCTCRGCCAATGGCGAGC	145 bp
		ConsORF2-a2	R: GTTCRTGYTGGTTRTCATAATCCTG	

ORF, Overlapping Open Reading Frame; Nucleotides = Y: T or C; R: A or G; K: G or T; W: A or T.

**Table 3 microorganisms-10-01628-t003:** Positive subjects reported basing on tissue types, age, sex and hunting geographical area.

Positive Samples N	Positive Tissues	Estimated Age and Sex	Hunting Area
1	LV; MS	AF	Comunanza (AP)
1	LV	AF	Funti (AP)
2	LV	AF	Vena Piccola (AP)
2	LV	AF	Venarotta (AP)
4	LV	PF	Vena Piccola (AP)
1	LV	PF	Comunanza (AP)
1	LV; MS	PM	Vena Piccola (AP)
1	LV	PM	Comunanza (AP)
2	LV; MS	AM	Vena Piccola (AP)
2	LV	AM	Roccafluvione (AP)

LV: Liver; MS: Muscle; A: Adult; P: Puberal; M: Male; F: Female.

**Table 4 microorganisms-10-01628-t004:** Positive animals to the HEV RNA detection classified basing on sex and tissue.

Sex and Liver Positivity		Sex and Muscle Positivity	
11 F (65.0%)	5 P (45.45%) 6 A (54.55%)	1 F (25.0%)	1 A (100.0%)
6 M (35.0%)	2 P (33.33%) 4 A (66.67%)	3 M (75.0%)	1 P (33.33%) 2 A (66.67%)
Total: 17 animals		Total: 4 animals	

M: Male; F: Female. J: Juvenile (weight < 15 kg and estimated age between 0–12 months). P: Puberal (15 kg < weight < 40 kg, and estimated age between 13–24 months). A: Adult (weight > 40 kg and estimated age between 24–48 months).

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
