# Peer review of "Hepatitis E Virus Detection in Hunted Wild Boar Liver and Muscle Tissues in Central Italy"

_microorganisms, 2022, doi:10.3390/microorganisms10081628_

Round 1

Reviewer 1 Report

Dear editor 

The objective of Manuscript ID microorganisms-1850990, entitled “Hepatitis E virus detection in hunted wild boar liver and muscle tissues in Central Italy’’ deals with an interesting topic and fits with the scope of the journal. 

Hepatitis E virus (HEV) has been recognized as emerging public health concern. The current study provides new data regarding HEV genotype and subtype circulation in wild boars in Italy. 

Generally, the current article is a well-designed and well-written work, providing novel results.

Therefore, I suggest that the article be accepted for publication, under minor revision.

Please let me congratulate you on the quality of your journal and thank you for giving me the opportunity to contribute as a reviewer.

Comments to the authors

§  L61: add more references 

§  L110-119: add appropriate reference 

§  L194-199: merge these three paragraphs in one, You could also write more 

§  Please provide Ethical Note for your study, including the approval number by the Ethics Committee 

§  Discussion: you should underline the significance of your results for public health in an individual  paragraph 

Author Response

Dear Reviewer 1,

All Authors want to express their appreciation to consider our manuscript for publication. Reviewers’ suggestions resulted precious and important to improve its quality and their observation were useful to implement the Scientific impact of our paper.

In the following two paragraphs we reported all changes:

Reviewer 1 suggestions:

Line 61: add more references.

Authors: Authors added further references (line 61), following reviewer suggestion.

Lines 110-119: add appropriate reference.

Authors: Authors added a reference [15], following reviewer suggestion.

Lines 194-199: merge these three paragraphs in one, You could also write more.

Authors: According to reviewer suggestion lines 194-199 were assembled and further considerations were reported from line 202 to 207.

Reviewer suggestion: Please provide Ethical Note for your study, including the approval number by the Ethics Committee.

Authors: It is not applicable. The screened tissues were provided by hunters to the Sanitary Authorities in accordance with Regional Marche Law No. 3/2012, because these animals were included in the Regional Hunting Program.

Reviewer suggestion on Discussion Section: You should underline the significance of your results for public health in an individual paragraph.

Authors: the significance of our results for public health in an individual paragraph has been added from line 301 to 307. In the line 305, authors added a further reference [31].

We really appreciate all provided observation and hope that our manuscript has increased its scientific values.

Thank You for Your time and attention.

Best regards,

Gianluigi Ferri

Doctor in Veterinary Medicine (D.V.M.)

Ph.D. Student in Food Inspection

Faculty of Veterinary Medicine; University of Teramo, Italy.

Reviewer 2 Report

The article describes hepatitis E virus prevalence in hunted wild boar liver and muscle samples in Central Italy. Research experiments were correctly planned.  Already published PCR primer sets used for the research give the research credibility. The research is described clearly, and the obtained results are beyond doubt.

I have no comments on the article. The information gathered is presented obviously, but due to the narrow experimental part, I suggest changing the type of the article to a short Communication type.

A proofreading mistake has been noticed in the text: lanes 172 and 174 should be Figure 2 instead of Figure 1.

Author Response

Dear Reviewer 2,

All Authors want to express their appreciation to consider our manuscript for publication. Reviewers’ suggestions resulted precious and important to improve its quality and their observation were useful to implement the Scientific impact of our paper.

In the following two paragraphs we reported all changes:

Reviewer 2 suggestions:

Reviewer suggestion: A proofreading mistake has been noticed in the text: lanes 172 and 174 should be Figure 2 instead of Figure 1.

Authors: Figure 2 instead of Figure 1 has been changed following reviewer suggestion.

Reviewer suggestion: I have no comments on the article. The information gathered is presented obviously, but due to the narrow experimental part, I suggest changing the type of the article to a short Communication type.

Authors: we really appreciate Your considerations. If it were possible, we would be pleasured to apport further changes to implement paper quality.

However, we suggest and believe that our manuscript should be considered as Original Article. This research study is a relevant part of our Faculty project. In this paper, authors designed the Scientific question in order to provide to a multidisciplinary perspective about hepatitis E virus circulation and its possible sanitary repercussions on consumers health. Therefore, we kindly suggest considering our paper as Original Article.

We really appreciate all provided observations and hope that our manuscript has increased its scientific values.

Thank You for Your time and attention.

Best regards,

Gianluigi Ferri

Doctor in Veterinary Medicine (D.V.M.)

Ph.D. Student in Food Inspection

Faculty of Veterinary Medicine; University of Teramo, Italy.
